# Preclinical evaluation of manufacturable SARS-CoV-2 spike virus-like particles produced in Chinese Hamster Ovary cells

Sergio P. Alpuche-Lazcano [1], Matthew Stuible[1], Bassel Akache[2], Anh Tran [2], John Kelly[3], Sabahudin Hrapovic[4], Anna Robotham[3], Arsalan Haqqani[3], Alexandra Star [3], Tyler M. Renner[2], Julie Blouin[1], Jean-Sébastien Maltais[1], Brian Cass[1], Kai Cui[5], Jae-Young Cho[5], Xinyu Wang[5], Daria Zoubchenok[1], Renu Dudani[2], Diana Duque[2], Michael J. McCluskie [2] & Yves Durocher [1✉]

## Abstract

**Background** As the COVID-19 pandemic continues to evolve, novel vaccines need to be developed that are readily manufacturable and provide clinical efficacy against emerging SARS-CoV-2 variants. Virus-like particles (VLPs) presenting the spike antigen at their surface offer remarkable benefits over other vaccine antigen formats; however, current SARS-CoV-2 VLP vaccines candidates in clinical development suffer from challenges including low volumetric productivity, poor spike antigen density, expression platform-driven divergent protein glycosylation and complex upstream/downstream processing requirements. Despite their extensive use for therapeutic protein manufacturing and proven ability to produce enveloped VLPs, Chinese Hamster Ovary (CHO) cells are rarely used for the commercial production of VLP-based vaccines.

**Methods** Using CHO cells, we aimed to produce VLPs displaying the full-length SARS-CoV-2 spike. Affinity chromatography was used to capture VLPs released in the culture medium from engineered CHO cells expressing spike. The structure, protein content, and glycosylation of spikes in VLPs were characterized by several biochemical and biophysical methods. In vivo, the generation of neutralizing antibodies and protection against SARS-CoV-2 infection was tested in mouse and hamster models.

**Results** We demonstrate that spike overexpression in CHO cells is sufficient by itself to generate high VLP titers. These VLPs are evocative of the native virus but with at least threefold higher spike density. In vivo, purified VLPs elicit strong humoral and cellular immunity at nanogram dose levels which grant protection against SARS-CoV-2 infection.

**Conclusions** Our results show that CHO cells are amenable to efficient manufacturing of high titers of a potently immunogenic spike protein-based VLP vaccine antigen.

## Plain language summary

Virus-like particles (VLPs) have a structure that is similar to viruses but they cannot cause infection or illness. If VLPs are injected into the body they produce an immune response similar to that seen following infection by a virus. This means that VLPs can be used as vaccines against viruses that cause illness in people. Many drugs, named biologics, are manufactured using living cells, including cells that were originally derived from Chinese Hamster Ovaries (CHO cells). We developed a simple method to produce VLPs similar to the SARS-CoV-2 virus in CHO cells. We show that vaccination of rodents with these VLPs prevents them from becoming ill following infection with SARS-CoV-2. These VLPs could become a part of an alternative, easily produced vaccine for the prevention of COVID-19 in humans.

[1] Human Health Therapeutics Research Centre, National Research Council Canada, 6100 Royalmount Avenue, Montreal, QC H4P 2R2, Canada. [2] Human Health Therapeutics Research Centre, National Research Council Canada, 1200 Montreal Road, Ottawa, ON K1A 0R6, Canada. [3] Human Health Therapeutics Research Centre, National Research Council Canada, 100 Sussex Dr, Ottawa, ON K1A 0R6, Canada. [4] Aquatic and Crop Resources Development Research Centre, National Research Council Canada, 6100 Royalmount Avenue, Montreal, QC H4P 2R2, Canada. [5] Nanotechnology Research Centre, National Research Council Canada, 11421 Saskatchewan Drive, Edmonton, AB T6G 2M9, Canada. ✉email: yves.durocher@cnrc-nrc.gc.ca

In pursuit of safe, economical, and effective vaccines against severe acute respiratory syndrome coronavirus 2 (SARS-CoV-2), an array of antigen production and delivery strategies has been explored. In USA and Canada, the first approved vaccines were based on mRNA- or adenoviral vector-induced expression of spike (S) antigen by host cells[1]. Purified and adjuvanted antigen vaccines, which can consist of S proteins (full-length or fragments) such as Nuvaxovid (Novavax) and VidPrevtyn (Sanofi/GSK), or S-decorated VLPs like Covifenz (Medicago), were slower to develop but have gained prominence with recent approvals by regulatory authorities. In this regard, Novavax NVX-COV2373, containing the ancestral S sequence, has shown after three doses to generate more robust geometric mean titers (GMT) of neutralizing antibodies against the Omicron BA.1 sublineage than the BNT162b mRNA vaccine[2]. Similarly, compared to the original Comirnaty (Pfizer BioNTech), booster doses of VidPrevtyn in different clinical trials were shown to restore immunity against different SARS-CoV-2 variants with higher neutralizing activity against Omicron BA.1[3]. VLPs displaying S protein are structurally reminiscent of wild-type viruses[4], promoting high immunogenicity due to their repetitive array of antigens which induces strong Th1/Th2 responses. Additional advantages of VLPs as vaccine antigens include storage stability, strong recognition and uptake by peripheral antigen-presenting cells (APCs); VLPs also eliminate the risk of revertant strains and the potential loss of key neutralizing epitopes associated with attenuated and inactivated viruses[5–8].

VLPs displaying S antigens are increasingly well-regarded as candidates for clinical SARS-CoV-2 vaccine development. VBI vaccines have completed Phase I trials of a vaccine based on VLPs based on retroviral Gag core protein co-expressed with S protein[9], a common approach for inducing VLP formation in mammalian cells. Medicago (Covifenz) achieved regulatory approval in Canada for VLPs produced in *N. benthamiana* plants by expression of a chimeric construct consisting of the S ectodomain fused to the transmembrane and C-terminal domains of influenza HA[10,11]. VLPs can also be produced by co-expression of the coronavirus structural proteins membrane (M), envelope (E), nucleocapsid (N), and S, an approach that results in VLPs more closely resembling the natural virion and which is also undergoing clinical evaluation[4,12].

The promise of VLPs as SARS-CoV-2 vaccine antigens is counterbalanced by common pitfalls including manufacturability challenges, poor particle yields, low S protein density on particles, and heterologous expression platform-driven non-human-like glycosylation. For these reasons, and because of our success at expressing the S protein at very high levels in Chinese Hamster Ovary (CHO) cells[13–15], we explored whether these cells, the predominant mammalian cell host for therapeutic protein manufacturing, could also be used to effectively produce SARS-CoV-2 VLPs.

Herein, we describe how the expression of native full-length SARS-CoV-2 S protein in CHO cells induces the release of abundant S-coated VLPs (S-VLP) without requiring the co-expression of any additional viral structural proteins. These S-VLPs can be produced in CHO cells engineered for abrogated endogenous retrovirus-like particle (ERVLP) production, which could otherwise contaminate the vaccine candidate. Effective purification of S-VLPs was achieved from cell culture supernatants using a one-step affinity chromatography protocol. A variety of biochemical and biophysical assays were performed to characterize these particles in detail, including S protein concentration, particle size, S surface density, and N-glycans identity. Finally, we demonstrated that purified S-VLPs are very potent as vaccine antigens in pre-clinical rodent models, providing effective protection at sub-microgram dose levels with licensed adjuvants.

We believe that their good manufacturability, stability, and efficacy features make these VLPs promising antigens for next-generation COVID-19 vaccines.

## Methods

**Cells and culture conditions**. NRC CHO²³⁵³™ cell lines have been described previously[16,17]. CHO²³⁵³ is a clone derived from CHO-DXB11 cells which were selected based on its high production yields of recombinant proteins. A complete description of the origin and selection of this clone can be consulted in ref. [15]. Clone CHO-C2 was derived from the parental CHO²³⁵³ by targeting and disrupting retroviral *gag* and *env* sequences responsible for ERVLP formation and release using CRISPR-Cas9 technology similarly as described in ref. [18]. Cells were maintained in a proprietary media formulation in polycarbonate Erlenmeyer flasks with 0.2 μm vent cap (Corning) with constant orbital shaking at 120 rpm in a humidified incubator with 5% $CO_2$ at 37 °C. All cells used in this study were negative for Mycoplasma.

**Plasmid constructs**. The coding sequences for SARS-CoV-2 S, M, and E were obtained from the genome ancestral sequence NC_045512. All S protein sequences contained mutations at the furin cleavage site (R682G, R683G, R685S), stabilizing prolines (K986P, V987P), and C-terminal HA epitope-tag (YPYDVP-DYA). Expression constructs for spikes were prepared by re-synthesizing and replacing restriction fragments encompassing mutations present in Beta B.1.351 (L18F D80A, D215G, L241del, L242del, A243del, R246I, K417N, E484K, N501Y, D614G, A701V), Delta B.1.617.2 (T19R, G142D, E156del, F157del, R158G, L452R, T478K, D614G, P681R, D950N) and Omicron B.1.1.529 (A67V, H69del, V70del, T95I, G142D, V143del, Y144del, Y145del, N211del, L212I, E215del, E216del, E217del, F339D, S371L, S373P, S375F, K417N, N440K, G446S, S477N, T478K, E484A, Q493K, G496S, Q498R, N501Y, Y505H, T547K, D614G, H655Y, N679K, P681H, N764K, D796Y, N856K, Q954H, N969K, L981F) variants. The SmT2v3 construct consists of the ectodomain of the SARS-CoV-2 ancestral strain S fused to T4 foldon and is identical to the ECDm-T4-Fib construct previously described[13] but without C-terminal epitope tags. All gene sequences were synthesized by GenScript, Piscataway, NJ, USA, and sub-cloned in pTT5® plasmid between EcoRI and BamHI restriction sites. Plasmids were transformed in E. *coli* DH5α bacteria (Invitrogen) and amplified in CircleGrow medium (MP Biomedicals) supplemented with 100 μg/mL ampicillin (Gibco). Plasmids were purified using a proprietary anion exchange chromatography method followed by isopropanol precipitation, ethanol wash, and 0.22 μm filtration. DNA quantification was carried out using a DeNovix DS-11+ spectrophotometer. Sequences were verified by Sanger sequencing using an ABI 3500xl genetic analyzer.

**Cell transfections**. CHO cell transfection methods were performed as described in ref. [13]. Individual structural SARS-CoV-2 gene transfections were performed in CHO media at 85% (w:w) of pTT5-Spike-HA, 10% Bcl-$X_L$ plasmid (anti-apoptotic effector) and 5% GFP plasmid. Co-transfection of structural SARS-CoV-2 genes was performed using a mixture of plasmids comprising 42.5% pTT5-membrane-envelope (M-E), 42.5% pTT5-Spike-HA, 10% Bcl-$X_L$ and 5% GFP diluted in CHO complete media.

**SARS-CoV-2 VLP harvesting and sedimentation**. Five days post-transfection, cell suspensions (viability ≥95%) were centrifuged for 15 min at 3200 rpm on a Sorvall legend RT centrifuge (Thermo Fisher Scientific). Cell supernatants were then treated with 10 U/mL of Denarase (C-LEcta), 5 mM of $MgCl_2$, and

incubated for 1 h at 37 °C. For VLP sedimentation, treated supernatants were centrifuged over a 15% Optiprep (w/v) (Sigma-Aldrich)[19] cushion (10% of total volume) at $5300 \times g$ for 16 h at 4 °C. After centrifugation, supernatants were discarded and pellets resuspended in DPBS (Cytiva).

**SARS-CoV-2 VLP and S purification by affinity chromatography.** Denarase-treated CHO-C2 supernatants were directly purified on AVIPure-COV2S VLP (AVIpure) resin (Avitide) packed in a BioRad PolyPrep chromatography column. AVIpure column was first equilibrated with 5 column volumes (CV) of DPBS. Next, the treated supernatant was loaded at a constant flow rate of 1 mL/min. The column was then washed with 5 CVs of DPBS. VLP was eluted with 50 mM Bis-Tris, 1 M $MgCl_2$ pH 6.0 followed by desalting into DPBS using NAP25 columns. Desalted VLPs were filter-sterilized and tested for endotoxins (Endosafe-PTS cartridges, Charles River) prior to in vivo experiments. The SmT2v3 soluble S ectodomain was produced from stable CHO cells as described previously[14] and purified using the NGL COVID-19 S Affinity resin (Repligen) following the manufacturer's instructions.

**Total protein quantification.** Sedimented or purified VLPs were lysed with RIPA buffer (150 mM NaCl, 1% NP-40, 0.5% sodium deoxycholate, 0.1% SDS, 50 mM Tris–HCl pH 7.4) supplemented with protease inhibitor cocktail (cOmplete, EDTA-free, Roche). VLP samples in RIPA buffer were incubated at 4 °C for 20 min followed by 15 min centrifugation at top speed in a benchtop centrifuge (Eppendorf 5427R). Pellets were discarded and supernatants recovered. Protein quantification was performed with Pierce bicinchoninic acid (BCA) protein assay (Thermo Fisher Scientific). In brief, 200 μL of BCA working reagent prepared according to the manufacturer's protocol was added to a 96-well plate and mixed with 25 μL of bovine serum albumin (BSA) standards (Thermo Fischer Scientific), sedimented or purified VLP lysates, and incubated for 30 min at 37 °C. Absorbance at 562 nm was measured using a SpectraMax 340PC spectrophotometer.

**SDS–PAGE and immunoblotting.** Following protein quantification, 0.7 μg of total protein from VLPs or 7.5 μL of Denarase-treated supernatants (unless otherwise indicated) were mixed (3:1 [v:v]) with XT sample buffer 4X (Bio-Rad) supplemented with 200 mM dithiothreitol (DTT), heat-denatured at 70 °C for 10 min under reducing conditions followed by separation of proteins by SDS–PAGE using NuPAGE 4–12% Bis–Tris gels. Total protein staining (Coomassie Blue) was performed using standard methods. VLP S protein concentration was determined by the comparison to an NRC Metrology soluble S trimer standard (STD) Reference Material SMT1-1[20] using a ChemiDoc MP imaging system (Bio-Rad) and Image Lab 6.1 software.

For immunoblotting, proteins separated by SDS–PAGE were transferred to nitrocellulose membranes using a Trans-Blot Turbo (Bio-Rad) apparatus with default settings. Membranes were blocked for 30 min with 5% milk in DPBS with 0.1% Tween 20 (DPBS-T) followed by three 5-min washes with DPBS-T. Membranes were then incubated for 1 h at room temperature with anti-S (S1) (ProSci #9083), anti-membrane (ProSci #3529), or a mouse anti-ERVLP-Gag p30 (produced in-house) at a 1:1000 dilution. Primary antibody incubation was followed by three 5-min washes with DPBS-T and incubation for 1 h with horseradish peroxidase-conjugated secondary antibodies at a 1/10,000 dilution for goat anti-rabbit (Jackson 111-035-003) and 1/5000 dilution for goat anti-mouse (Sigma A2304). After three additional 5-minute washes with DPBS-T, the bands were

visualized with Clarity Western ECL Substrate (Bio-Rad) and ChemiDoc MP imaging system. All uncropped gel/blot images can be consulted in Supplementary Fig 5.

**Detection of hACE2 binding of S protein on S-VLPs by ELISA.** The ELISA method to detect S-VLP binding to hACE2 is provided in the Supplementary Methods section.

**Host cell protein quantification.** CHO HCP quantification was determined by ELISA using the CHO Cygnus kit (Cat # F550-1) according to the manufacturer's protocol.

**Negative-stain transmission electron microscopy (TEM).** Observation of VLPs by transmission electron microscope (TEM, HITACHI H-7500 and HITACHI HT7700 120) equipped with bottom-mounted AMT NanoSprint 12MP camera and operating at 80 kV in high-contrast mode, was performed using negative staining. TEM grids (Cu 200 mesh, 15–25 nm carbon supported) were freshly glow-discharged using EMS GloQube-D, Dual chamber glow discharge system (Electron Microscopy Sciences) in negative mode with a plasma current of 25 mA during 45 s. Such grids were floated on 10 μL sample aliquots on Parafilm for 5 min. The excess droplets were subsequently wicked away from the edge of the grid with filter paper strips (Whatman 541). The grid was then rinsed three times with three droplets of double distilled water each time removing the excess. Immediately after the last rinse, the grid was exposed to the staining solution (1% uranyl formate) for 60 s and the stain was carefully removed using a fresh piece of filter paper. Finally, the grid was dried at room temperature for 2 h and used for TEM analysis.

**Cryo-electron microscopy (Cryo-EM).** S-VLP samples were prepared by cryo-plunge[21] with a Leica-EM GP2 plunger. TEM grids (Cu 400 mesh, with holey carbon film) were treated with a Harrick PDC-32G plasma cleaner with a plasma power of 18 W for 60 s. Such grids were picked up with adapter-attached forceps and loaded in an environmental chamber (humidity ~80%) of the Leica-EM GP2 unit. 5 μL of the aqueous sample was applied on the carbon-film side of the TEM grid and blotted with a filter paper (Whatman #1) for 1.5 s. The grid was then plunged into liquid ethane with a temperature of −180 °C and transferred to a cryo-sample box in liquid nitrogen (LN2). The sample box was further transferred in LN2 and loaded to a pre-cooled (−180 °C) Gatan 914 cryo-specimen holder in a workstation filled with LN2. The holder with the cryo-grid protected under an anti-frost cover was inserted in the TEM column. Observation of the cryo-plunged VLPs sample was carried out on a JEOL 2200 FS TEM, operating at 200 kV, equipped with a field emission filament. Image collections were under low dose mode[22] to mitigate beam damage. A 10 eV wide energy filter (with only zero energy loss electrons contributing to image formation) was used to enhance image contrast. Each image was collected with an exposure time between 0.5–1 s based on the local vitreous ice thickness and the microscope settings. Images were processed with the built-in functions (smooth, contrast reversal) of Gatan DigitalMicrograph software.

**S quantification and model building of S-VLP.** S protein quantification at the VLP surface and modeling method are described in the Supplementary Methods section.

**Nanoparticle tracking analysis (NTA).** S-VLPs purified material and mock samples contained in DPBS were analyzed by NTA with a NanoSight NS 5000 (Malvern Panalytical) equipped with a laser wavelength of 532 nm and a 565 nm long-pass filter. Two

sets of data were acquired. Parameters for NTA measurements of samples were performed as follows: VLPs and polystyrene beads were captured 3 times (1 independent preparation) at a temperature of 20 °C with the camera Level at 16 (Slider Shutter 1300; Slider Gain 512). The following analysis settings were used to process the acquired data; Detection Threshold 3; Blur Size and Max Jump Distance were Auto. VLP samples were diluted in DPBS at a minimal final volume of 1 mL to allow replicate injection. Aspiration of the sample into the system was done using the integrated peristaltic pump connected to the tubing. A minimal volume of 1.8 mL was prepared for each sample to allow technical replicate analysis. NanoSight NS 5000 calibration was carried out by loading 100 nm polystyrene beads, 1/100,000 (Thermo Fisher Scientific) with a predetermined acceptance criteria of 15–90 particles per frame and a ≤15% of CV between injections from the same preparation.

Mock samples were diluted in DPBS at 1:475. S-VLPs were diluted at 1:500, 1:1000, 1:1250, 1:1500, and 1:2000. Image processing, particle size distribution, and concentration calculations were performed with NanoSight NTA 3.2 Analytical Software (Malvern Panalytical) with a finite track length adjustment (FTLA) fitting model.

**N-glycosylation of S protein in VLPs and soluble S (SmT2v3).** S protein glycosylation analysis method for S-VLPs or S protein is described in detail in the Supplementary Methods section.

**Mice and hamsters.** Female C57BL/6 mice (6–8 weeks old) and Syrian golden hamsters (81–90 g) were purchased from Charles River Laboratories (Saint-Constant, Canada). Animals were maintained at the animal facility of the National Research Council Canada (NRC) in accordance with the guidelines of the Canadian Council on Animal Care. All procedures performed on animals in this study were approved by our Institutional Review Board (NRC Human Health Therapeutics Animal Care Committee) and covered under animal use protocol 2020.06 and 2020.10. All experiments were carried out in accordance with the ARRIVE guidelines. Findings in the present manuscript do not apply to a particular sex, but female rodents were preferred because they tend to be less aggressive during manipulation and socialize better between them.

**Animal immunization and sample collection.** Eight groups ($n = 10$ per group) of mice and four groups ($n = 6$ per group) of hamsters were immunized with different formulations to evaluate the immunogenicity of S-VLPs. In brief, affinity-purified S-VLPs and adjuvant vaccine components were admixed and diluted in PBS (Thermo Fisher Scientific) prior to administration in a final volume of 50 µL per dose. Adju-phos (Invivogen) dose levels were based on data from previous studies with 50 µg $Al^{3+}$ included per dose[23]. AS01b (GlaxoSmithKline, Brentford, UK) based on the saponin QS-21 and TLR4 agonist monophosphoryl lipid A (MPL) was used at 1/20th of the human dose.

Animals were immunized by intramuscular (i.m.) injection (50 µL) into the left tibialis anterior (T.A.) muscle on days 0 and 21. Vaccine formulations of 3 µg of S-VLPs alone, or 3, 0.3, and 0.06 µg of adjuvanted S-VLPs were administrated to mice. Vaccine formulations for hamster studies consisted of: PBS vehicle (naïve), 0.3 µg of S-VLPs, AS01b alone, and 0.3 µg S-VLPs +AS01b. S-VLP quantities are based on the amount of S protein present in the purified particles and adjuvant amounts were kept constant regardless of antigen dose. On day 28, mice were anesthetized with isoflurane and then euthanized by cervical dislocation prior to the collection of spleens for measurement of cellular immune responses by IFN-γ ELISpot. Mice were bled via the submandibular vein on days 20 and 28 with recovered serum used for quantification of antigen-specific IgG antibody levels and neutralization assays. On day 35 hamsters were challenged intranasally by $1 \times 10^5$ PFU of SARS-CoV-2 ancestral strain (Canada/ON/VIDO-01/2020, obtained from the National Microbiology Lab, Winnipeg, Canada) and monitored for body weight change for the next five consecutive days. On day 40, animals were euthanized by exposure to $CO_2$ (following anesthesia by isofurane) and lung tissues were collected for assessment of viral load by plaque assay and viral RNA detection and quantification by RT-qPCR.

**Mouse serum anti-S ELISA.** The method to determine total anti-S IgGs has been described previously[23] and can be consulted in detail in the Supplementary Methods section.

**Cell-based surrogate SARS-CoV-2 neutralization assay.** The surrogate neutralization assay was performed as in ref. [24] and is described in the Supplementary Methods section.

**Viral RNA detection by RT-qPCR.** Viral RNA (vRNA) was isolated from lung tissue homogenized in 1 mL RNA/DNA Shield (Zymo Research). vRNA was extracted under BSL-3 conditions with Quick-RNA Viral Kit (Zymo Research). Viral genomic RNA was quantified by real-time-PCR to the target viral envelope gene as previously described[25].

**Plaque assay.** This assay was carried out in containment level 3 facilities (CL3) as previously described[23]. Briefly, the whole left lungs from each hamster were individually homogenized in 1 mL PBS and centrifuged. The clarified supernatants were then diluted 1 in 10 serial dilutions in infection media (1×DMEM, high glucose media supplemented with 1×non-essential amino acid, 100 U/mL penicillin–streptomycin, 1 mM sodium pyruvate, and 0.1% bovine serum albumin). Next, Vero E6 cells were infected for 1 h at 37 °C before the inoculum was removed and overlay media was added, which consisted of infection media with 0.6% ultrapure, low-melting point agarose. Infected cells were incubated at 37 °C with 5% of $CO_2$ for 72 h. After incubation time, cells were fixed with 10% formaldehyde and stained with crystal violet. Plaques were enumerated and PFU was determined per gram of lung tissue.

**Statistics and reproducibility.** All graphs and statistical analyses in the figures presented in this manuscript were performed by GraphPad Prism 9.3.1 (GraphPad Software). Eight groups ($n = 10$ per group) of mice and four groups ($n = 6$ per group) of hamsters were assessed for in vivo experiments. A one-way ANOVA with Tukey's multiple comparison tests were performed to assess the significance of total anti-S IgG and quantification of spike-specific T cells analyses. A one-way ANOVA with Dunnette's multiple comparison tests was performed to assess the significance of cell-based surrogate neutralization assays, virus quantification by plaque assay, and viral RNA quantification in hamster lungs. $p \leq 0.05$ was regarded as statistically significant.

**Reporting summary.** Further information on research design is available in the Nature Portfolio Reporting Summary linked to this article.

## Results and discussion
**SARS-CoV-2 S expression in CHO cells is sufficient to generate and release VLPs in cell supernatants.** Given the advantages and widespread use of CHO cells for therapeutic protein manufacturing[26], we initially explored their potential for

producing VLPs by co-transfection of the SARS-CoV-2 structural proteins S, M, and E (Supplementary Fig. 1a). These have been reported to be the minimal viral components allowing for the formation of VLPs when co-expressed[27,28]. At 5 days post-transfection, we observed the presence of M and S proteins in sedimentable particles from cell supernatants by SDS–PAGE (total protein staining and western blotting, Fig. 1a). However, negative-stain transmission electron microscopy (TEM) imaging detected few enveloped particles and those that were observed did not harbor a well-defined corona, a characteristic of coronaviruses, as the density of protruding S proteins was quite low (Fig. 1b).

Remarkably, when S was expressed alone, substantially higher levels of this protein were detected in CHO cell supernatants despite the presence of an intact transmembrane domain that should prevent its secretion as a soluble protein. S protein could also be sedimented by ultracentrifugation on an iodixanol cushion (Fig. 1a), suggesting that S alone triggered the formation of VLPs. Strikingly, we observed by TEM the presence of a large quantity of generally spherical VLPs surrounded by a well-defined corona formed by a dense coat of S protein (Fig. 1c). The ability to form VLPs using this approach was not limited to the ancestral S sequence: transfection of CHO cells with expression plasmid constructs encoding S proteins from Beta, Delta and Omicron variants also generated VLPs. Upon expression of these constructs, we detected the presence of S proteins in iodixanol-sedimented samples by western blot (Fig. 1d) and confirmed the formation of VLPs from different SARS-CoV-2 S variants by TEM (Fig. 1e). The latter demonstrates that this approach can be applied to different S variant sequences. The remaining experiments, including purification, characterization and in vivo testing were performed using VLPs displaying S of the ancestral strain. To the best of our knowledge, this is the first report describing the production of SARS-CoV-2 VLPs from mammalian cells by overexpression of the native S protein without any other viral components.

**Purification of SARS-CoV-2 S-VLPs produced by ERVLP-KO CHO cells**. The release of ERVLPs by CHO cells[29,30] could be a concern for recombinant enveloped VLP vaccine manufacturing, in particular, because of the likely structural and biophysical similarity of recombinant enveloped VLPs and endogenous CHO ERVLPs that would make separation of the two entities challenging during downstream processing steps. CRISPR-Cas9 has been used to target retrovirus-like proviral elements like *gag* and *env* in the CHO genome[18], and we used this approach similarly to engineer a CHO cell line, CHO-C2, with disrupted ERVLP production. As shown in Fig. 2a, CHO-C2 gives S-VLP yields similar to non-engineered cells but without the presence ERVLPs, as indicated by Gag western blotting (Gag is the principal protein component responsible for the formation and budding of these particles)[18]. Volumetric productivity for S-VLPs produced in CHO-C2 cells (harvested at 5 days post-transfection) averaged 8.5 ± 0.4 mg/L (standard error of the mean [SEM]; $n = 6$), based on S protein content of iodixanol-sedimented particles (Fig. 2b). In literature, describing other methods to produce VLPs displaying SARS-CoV-2 S, volumetric titers of S protein in unpurified supernatants are rarely shown. In one report, Gag-based VLP supernatant from HEK293 cells was estimated to contain 1.78 mg/L of S protein[31]. It is noteworthy that the downstream purification process for Gag-S VLPs can drastically reduce S concentrations[31].

It is generally challenging to develop purification schemes to separate recombinant enveloped VLPs from ERVLPs and other extracellular vesicles spontaneously released from host cells. For

**Table 1 Representative CHO-C2 HCP determination by ELISA in purified S-VLP solutions.**

| Sample | CHO HCP (ng/mL) | S (ng/mL) | CHO HCP (ppm) |
|---|---|---|---|
| Purified S-VLPs | $2.46 \times 10^3$ | $5.22 \times 10^4$ | $4.72 \times 10^4$ |

S-VLP purification, we tested several conventional column chromatography resins with unsatisfactory results before proceeding to evaluate affinity chromatography options, including commercial S-affinity resins (e.g., Repligen or Avitide) and in-house preparations of beads conjugated to S antibodies or recombinant human ACE2 (hACE2). All of the tested resins effectively bound VLPs, but their elution under non-denaturing conditions was quite inefficient. To address this issue, an S affinity resin with lower ligand density was generated by Avitide (AVIPure COV2S VLP). This resin permits effective capture and gentle elution of S-VLPs, which could then be buffer-exchanged and sterilized by 0.2 μm filtration for in vivo testing. Compared to iodixanol sedimented samples, the affinity-purified S-VLPs contain visibly lower levels of non-S proteins by total protein staining and show a high particle concentration with preserved morphology by TEM (Fig. 2c–e). To evaluate whether the S protein on the VLP surface is able to bind its receptor hACE2 by the exposure of the receptor-binding domain (RBD), we performed a sandwich ELISA assay based on the capture of VLPs by recombinant hACE2 followed by detection with anti-spike antibodies. As shown in Fig. 2g, S-VLPs bind to hACE2-coated but not to control uncoated plates.

Using a CHO host cell protein (HCP) ELISA kit as an orthogonal assay to assess purity, we determined that one-step affinity-purified VLP preparations contain approximately $47 \times 10^3$ ppm HCPs (relative to total protein) (Table 1). For a prospective clinical dose of 3.75 μg of S protein (based on the Covifenz SARS-CoV-2 VLP vaccine[10]), this would correspond to 0.17 μg HCPs in S-VLPs. By comparison, the ChAdOx1 nCov19 vaccine (AstraZeneca) has been reported to contain substantially more HCPs per dose[32]. Our CHO-C2-based production method followed by one-step affinity purification yielded 2.4 ± 0.3 mg/L (SEM) of VLPs based on S protein content (Fig. 2g). Notably, other VLP platforms that rely on co-expression of Gag with S to drive VLP formation give purified ratios of S to Gag that is very low (e.g., purified VLPs described by VBI which contain at most 2.78% S protein relative to Gag[9]) with much lower total levels of S protein (0.015 mg of S in VLPs purified from one liter of HEK293 supernatant[9,31]). In such cases, Gag is by far the most abundant protein in the final, purified VLP product (compared to S-VLPs which are composed of >80% S protein); although Gag-S-based VLPs have been shown to be effective immunogens in several studies, we believe that the higher S density and reduced levels of non-relevant structural proteins are a likely advantage of our VLP platform.

We also established the quantity and size of VLPs by nanoparticle tracking analysis (NTA) and assessed stability during storage at 4 °C. S-VLPs purified from a 1 L culture contained $1.38 \times 10^{13}$ particles with a median width of 119–128 nm. In comparison, purified SARS-CoV-2 VLPs generated in insect cells by the expression of M + E + S or Gag-S VLPs from HEK293 cells yielded up to $5.8 \times 10^{11}$ and $6.03 \times 10^{10}$ particles per 1 L of culture respectively[31,33]. Of note, S-VLP particle concentrations remained constant during storage for at least 100 days at 4 °C, indicating very good storage stability (Table 2, Supplementary Fig. 2).

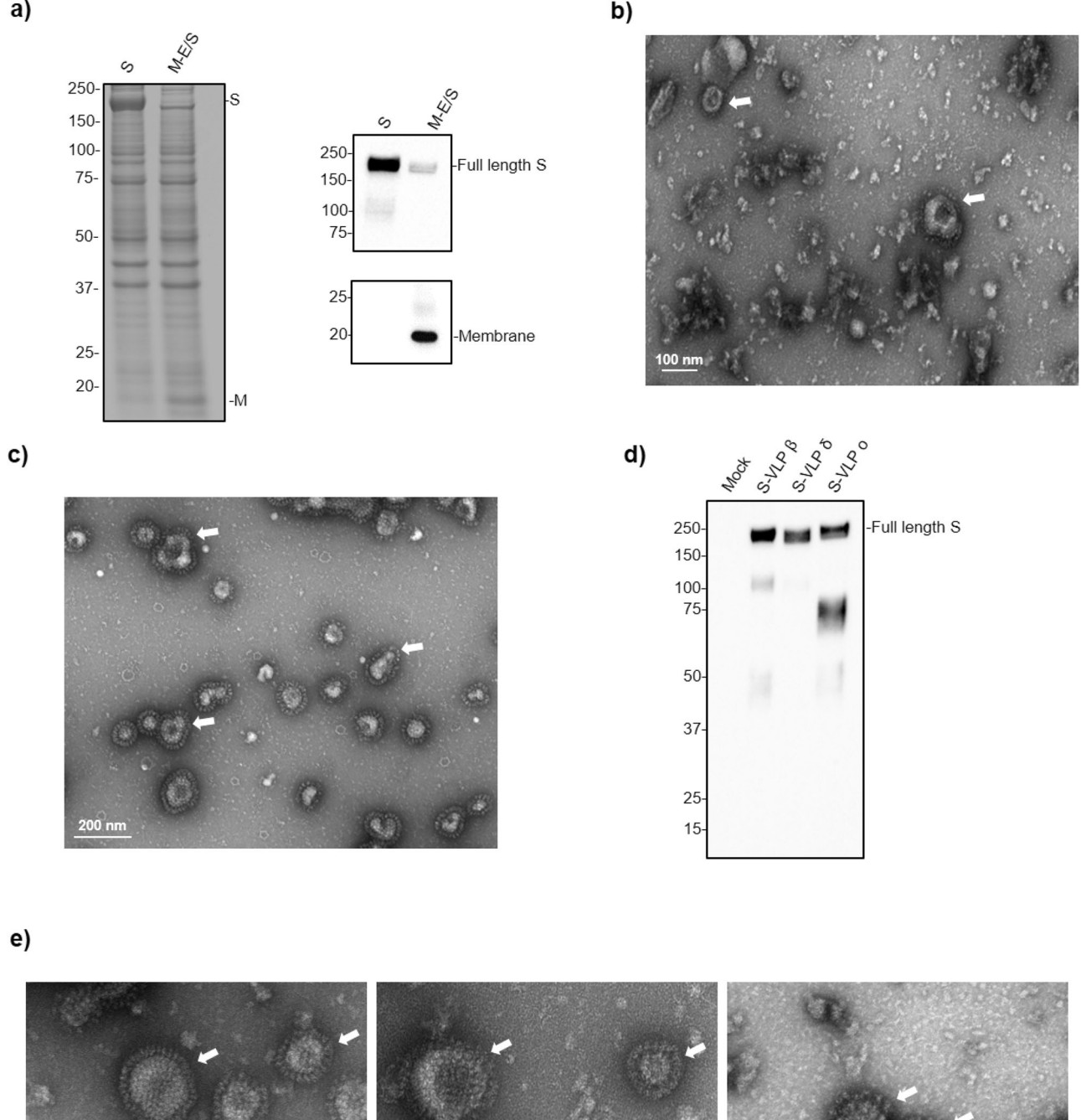

**Fig. 1 S expression in CHO cells is sufficient to generate S-coated SARS-CoV-2 VLPs. a** Sedimented supernatants with 15% iodixanol cushion from CHO[2353] cells transfected with S or M-E/S plasmids. On the left, total protein staining (Coomassie blue) of S or M-E/S. On the right, immunoblot with anti-Spike (S1) detecting S (~180 kDa). The SARS-CoV-2 membrane (M) protein is detected with anti-M at ~17 kDa. **b** Representative TEM image of sedimented M-E/S VLPs. White arrows indicate potential VLPs. A scale bar of 100 nm is shown at the bottom of the image. **c** Representative TEM image of sedimented VLPs generated by the expression of S protein (S-VLPs). A scale bar of 200 nm is shown at the bottom of the image. White arrows indicate VLPs. **d** Representative immunoblot of different S-VLPs variants detected with anti-S1. From left to right, mock, beta (B.1.351), delta (B.1.617.2), and omicron (B.1.1.529) S-VLPs. **e** Representative TEM images of sedimented S-VLPs variants. From left to right, beta (B.1.351), delta (B.1.617.2), and omicron (B.1.1.529). A 50 nm scale bar is shown at the bottom of each image.

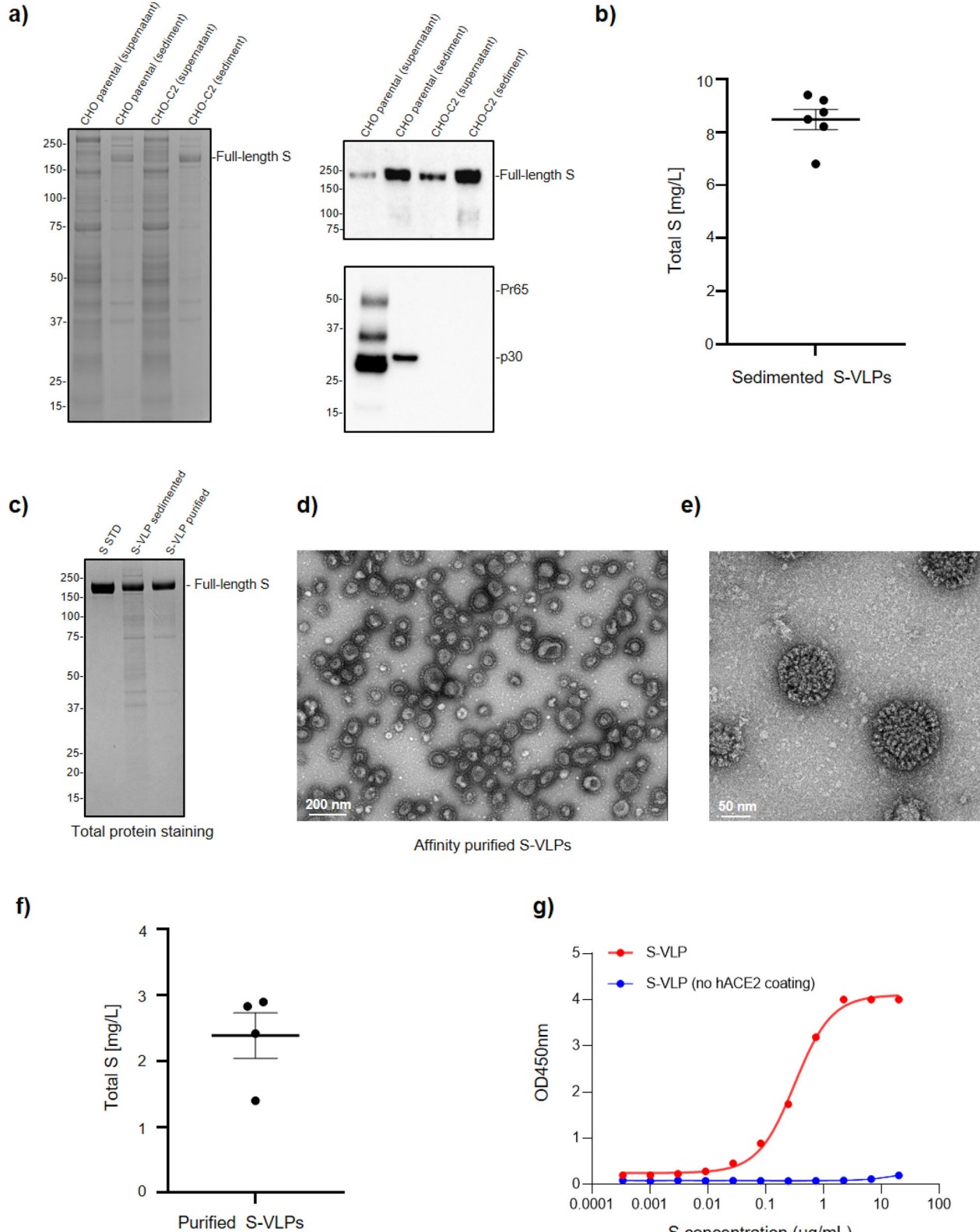

**Determination of S density and glycan structures on S-VLPs.** Immunization by intramuscular (i.m.) injection favors rapid recognition and uptake of vaccine antigens by peripheral APCs[34]. In this regard, antigen density on the pathogen or particle's surface along with its glycosylation pattern may impact immunogenicity and thus, the resulting immune response[5,35]. Negative stain TEM imaging indicates that S-VLPs contain more S on their surface (Fig. 2e) than SARS-CoV-2 virions produced in Vero E6 cells[36,37]. To investigate this further, we visualized particles by Cryo-EM. Peripheral S protein trimers on VLPs are reminiscent of the prefusion configuration with average lengths of ~20 nm (Fig. 3a), similar to those reported in previous studies[37]. The high resolution of the Cryo-EM images allowed us to count circumferential S protein trimers on a large number of VLPs with

**Fig. 2 S protein is the main component in purified S-VLPs derived from CHO-C2 cells. a** CHO-C2 (ERVLP-KO) cells. On the left, total protein staining (Coomassie blue) showing supernatants (7.5 µL) and sedimented (0.35 µg of total protein) VLPs produced in CHO[2353] (parental) or CHO-C2. On the right, Western blotting of VLPs produced by CHO[2353] or CHO-C2. The upper part of the blot shows the full-length S protein detected by anti-Spike (S1) in clarified harvest or sedimented VLPs. The lower blot is stained with anti-Gag p30 and shows the absence of Gag in CHO-C2-derived VLPs. **b** Scatter plot of total S protein (mg/L) from different samples at 5 days of transfection. Data are presented as mean ± SEM of six independent samples. **c** Total protein staining comparison of sedimented and affinity-purified S-VLPs. On the left, 1 µg of recombinant, soluble S protein (S STD) was loaded. For VLPs, 0.7 µg of total protein was loaded per well. **d** Representative TEM image of affinity-purified S-VLPs. A 200 nm scale bar is shown at the bottom. **e** Representative high-resolution TEM image of individual affinity-purified S-VLPs acquired with a HITACHI HT7700 120. A 50 nm scale bar is shown at the bottom. **f** Scatter plot of total S protein (mg/L) from different purified samples at 5 days of transfection. Data are presented as mean ± SEM of four independent samples. **g** ELISA assay. Dose-response curve of purified S-VLP (starting concentration [S] = 0.081 mg/mL) showing 11 serial dilutions. Red dots represent S-VLPs on a hACE2-coated plate in an S-shaped sigmoidal curve in red. Blue dots represent S-VLPs on an uncoated (mock) plate connected by a blue line.

**Table 2 NTA quantification on S-VLP purified samples over time.**

| Days | VLP particles/mL | Average size (nm) | Median size (nm) |
|---|---|---|---|
| 20 | $4.5 \times 10^{11}$ | 143 | 128 |
| 59 | $4.2 \times 10^{11}$ | 128 | 119 |
| 101 | $4.8 \times 10^{11}$ | 110 | 125 |

core diameters between 40 and 70 nm and establish a correlative function relating S number to VLP core diameter (Supplementary Fig. 3a). With a strong number/diameter ratio correlation between particles ($R^2 = 0.92$), we then calculated the real Euclidian distance which generated equidistant points between spikes (Fig. 3b). This allowed modeling S-VLPs three-dimensionally using MATLAB to estimate the total number of spikes for a VLP of a given size (e.g. VLP of 70 nm diameter contains 86 spikes) as shown in Fig. 3c. Based on this model we can confirm that overall, CHO-produced S-VLPs contain a higher S content than the average reported for intact SARS-CoV-2 virions ($25 \pm 12$ spikes)[36,37].

The capability of CHO cells to provide glycoproteins with human-like glycosylation could be advantageous for vaccine antigen candidates compared to those produced in non-mammalian cell hosts (e.g. plants, yeast, or insect cells)[38]. Glycan composition might not necessarily impact S protein structure or receptor binding but could drastically change its thermal stability or potency to generate neutralizing antibodies[39]. Importantly, non-stabilized S proteins (similar to the ones expressed by ChAdOx1 nCoV19) have shown different glycosylation patterns compared to stabilized recombinant trimeric S and S of SARS-CoV-2 virions all derived from human cells, which might also influence antigenicity and other biological properties[40].

To investigate possible glycosylation discrepancies between CHO-produced soluble S ectodomain and S-VLP, we conducted LC-HCD MS/MS assays, and profiled glycans present at 20 out of 22 potential N-glycosylation sites of the S protein present on S-VLPs or produced as a soluble, recombinant ectodomain trimer (SmT2v3), also from CHO cells. For both sample types, N-glycosylation sites were mostly sialylated and core-fucosylated with complex-type glycans which tended to be biantennary or triantennary. High mannose was commonly observed at specific sites and low levels of hybrid glycans were also observed at some other sites (Fig. 3d and Supplementary Fig. 3b). A major difference between soluble and VLP-S was detected at N1194. Glycosylation site N1194 on the VLP-S was glycosylated with putative N-acetyllactosamine (LacNAc) repeats which were not detected in SmT2v3, in which this site was mostly unoccupied (Fig. 3d). To the best of our knowledge, no reported biological function associated with the presence or absence of a glycan at N1194 exists. Overall, our analyses revealed that glycosylation of the VLP-associated and soluble S protein is quite

similar (Fig. 3d and Supplementary Fig. 3b) and are comparable to that found on soluble S protein produced by HEK293F cells as well as SARS-CoV-2 virion-associated S produced by Vero cells[36,41,42].

In contrast, S glycosylation can differ notably for proteins and VLPs produced in other production hosts used for vaccine manufacturing. *Nicotiana benthamiana* plants used for the production of VLPs by Medicago generate S with glycans known as putative plant allergens[43–45]. Also, insect cells such as Sf9, which are used to manufacture Novavax and Sanofi S-based vaccines, produce glycoproteins with primarily high mannose, oligomannose, and paucimannose glycans (the latter may also contain an immunogenic α1,3-linked core-fucose residue)[45–48]. More studies will be necessary to elucidate the impact of these glycan structures on the S protein immunogenicity. APC processing and presentation to B cells of an S protein containing immunogenic glycans or glycans not normally found on SARS-CoV-2 virions may divert immune response towards epitopes that do not protect against the natural virus.

Taken together, these results demonstrate that CHO-produced S-VLPs display a higher density of S proteins on their surface than the native virus produced in Vero cells with glycosylation similar to that reported previously for S produced by human and Vero cells (as soluble ectodomain or associated with virus particles).

**S-VLPs elicit protective humoral and cellular immune responses at nanogram dose levels.** The potential of S-VLPs as vaccine antigens, alone or in combination with adjuvants, was assessed in mouse and hamster in vivo models. Mice were first immunized at day 0 and then boosted at day 21 with VLPs alone (containing 3 µg S protein), or at different doses (3, 0.3, or 0.06 µg S protein) with Adju-phos or AS01b adjuvants. Analysis of day 28 serum samples demonstrated that VLPs induced anti-S antibodies even without adjuvant, but as low as 60 ng of VLPs induced titers >4-fold higher than the antigen alone when adjuvanted with Adju-phos and >23-fold higher with AS01b (Fig. 4a). IFN-γ ELISpot showed that specific T cell responses are elicited strongly by VLPs+AS01b at all doses tested, moderately by antigen alone but inhibited when combined with Adju-phos (Fig. 4b). A surrogate neutralization assay using ancestral-strain S protein[23] demonstrated substantial variability in neutralization activity with serum from the VLP and VLP+Adju-phos groups (Fig. 4c). Conversely, serum from the VLPs+AS01b group reached maximal neutralizing activity against reference-strain S, given the dilution used in our protocol. Beta, Delta, and Omicron BA.4-5S protein variants were also effectively neutralized by serum from mice immunized with VLPs+AS01b even with the lowest 60 ng dose tested (Supplementary Fig. 4). Based on these results, we conclude that VLPs+AS01b elicits a strong humoral and cellular immune response against the ancestral and different virus strains in mice. Conversely, VLPs and VLP+Adju-phos

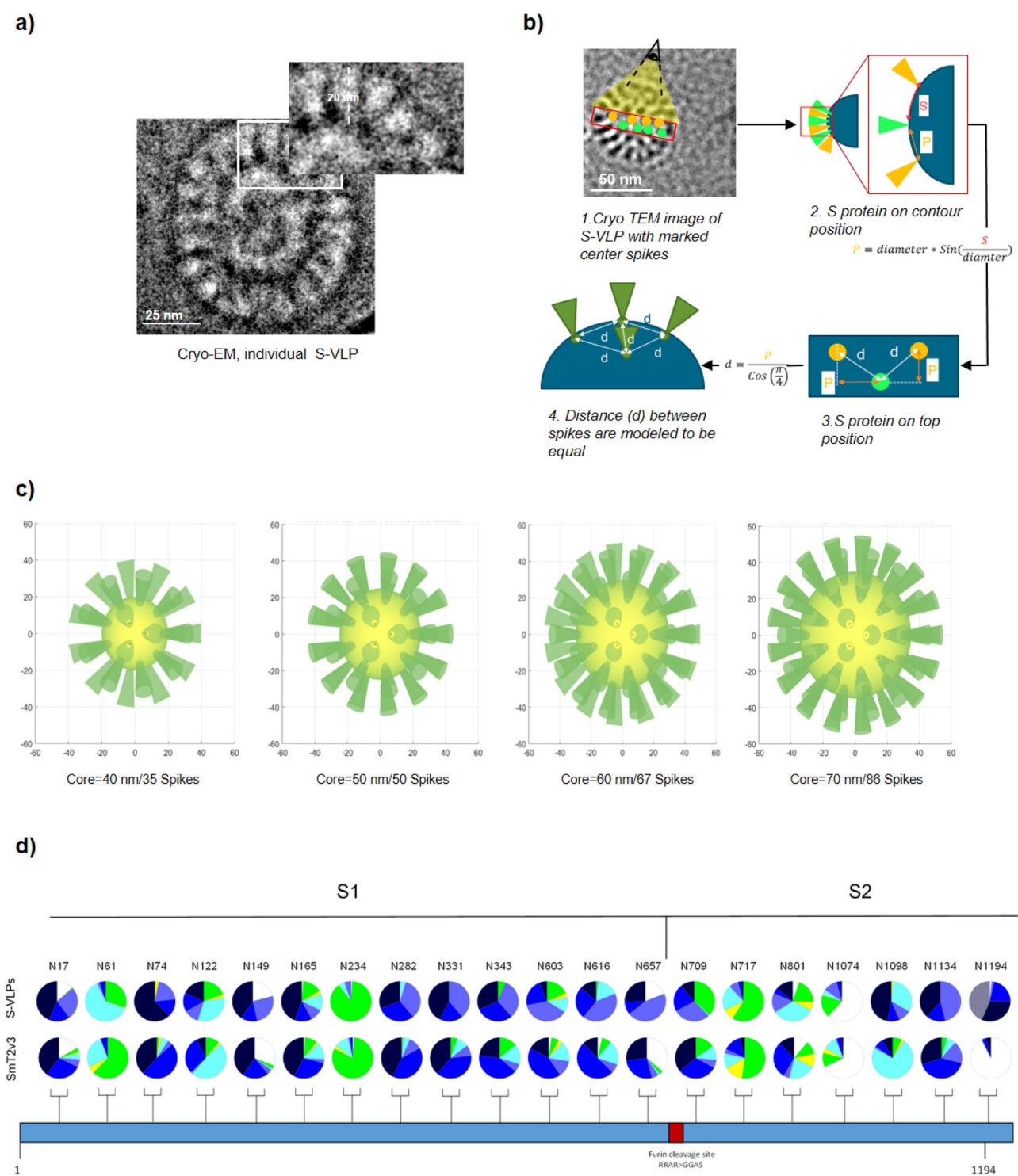

**Fig. 3 S-VLPs characterization shows a high number of S proteins on the particle surface with a glycan pattern similar to soluble S trimmers.**
**a** Individual S-VLP image (positive contrast) was acquired by Cryo-EM. The portrayed VLP contains 20 peripheral S proteins. Individual S proteins are evident in the magnified image, protruding from the envelope with an average size of 20 nm. A 25 nm scale bar is displayed at the bottom of each image. **b** Schematic of calculation workflow from the measured arc length between two peripheral spikes to real Euclidian distance between spikes on the VLP core surface. **c** Reconstruction of S distribution on VLPs using the real Euclidian distance. VLP models correspond to core diameters of 40 nm/35 spikes, 50 nm/50 spikes, 60 nm/67 spikes, and 70 nm/86 spikes. **d** Compositional glycan identification of the S-VLP and recombinant soluble (SmT2v3) S proteins. The identity and proportion of 20/22 N-glycans were determined by LC–MS and presented in pie charts. See also Supplementary Fig. 3 and Supplementary Data file.

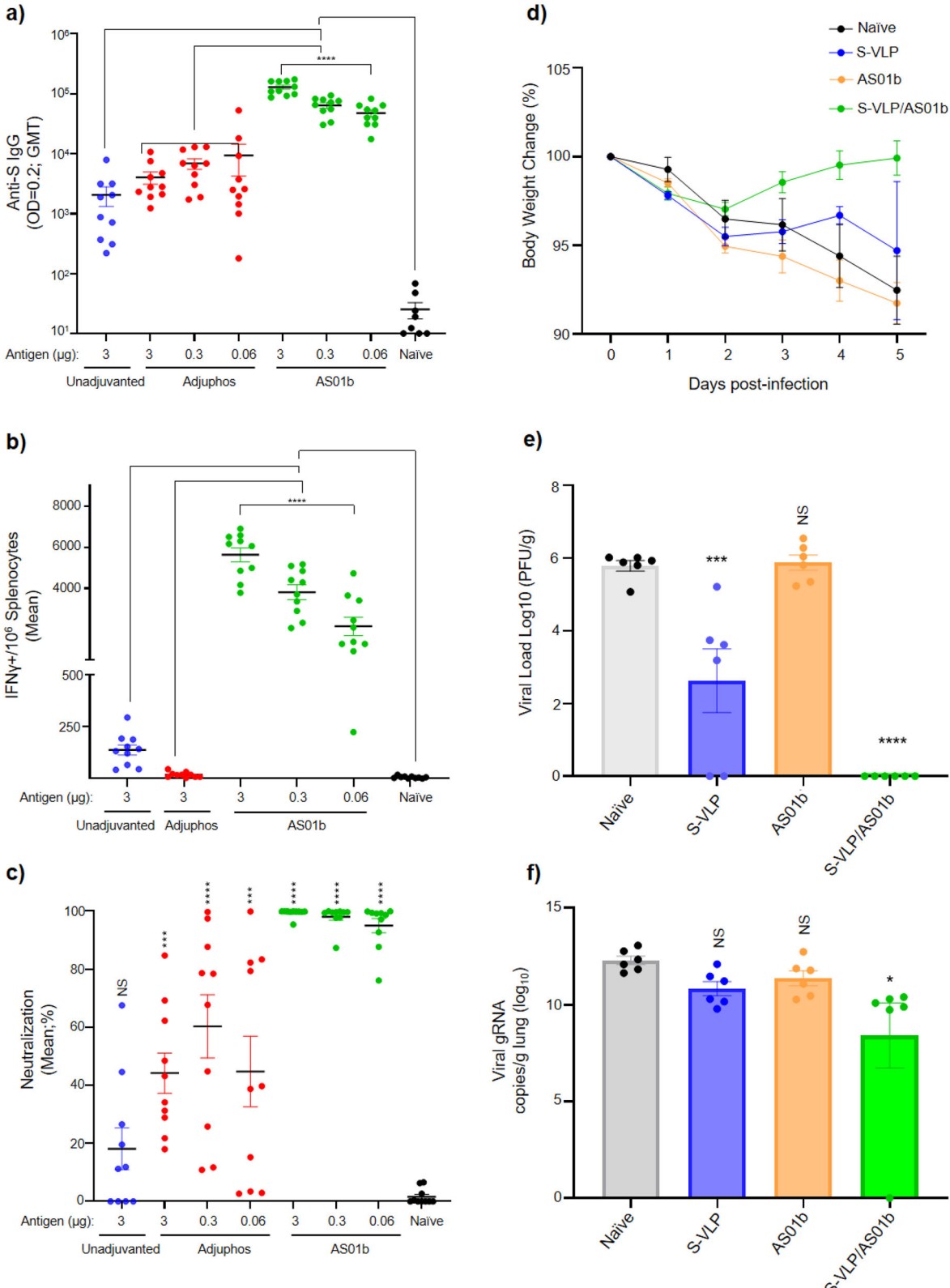

triggered specific IgGs but tended to be less immunogenic along with a null or less potent neutralizing activity. Importantly, VLP +Adju-phos seems to impair the cellular response (Fig. 4b). Therefore, we decided not to pursue further testing of this adjuvant.

We next evaluated protection provided by immunization with S-VLP formulations in a hamster challenge model. After two vaccine doses followed by infection with SARS-CoV-2, a marked reduction in weight loss was observed for the VLPs+AS01b group compared to non-vaccinated control and non-adjuvanted VLP groups (Fig. 4d). Correspondingly, a lung homogenate plaque assay showed dramatically reduced SARS-CoV-2 viral load in the VLPs + AS01b group and a smaller but notable reduction in animals that received unadjuvanted VLPs ($p = 0.0003$) (Fig. 4e).

**Fig. 4 Immunization with adjuvanted S-VLPs induces SARS-CoV-2-neutralizing, protective responses in rodents. a\*** Humoral immune response titers (total anti-spike IgG) in vaccinated mice with different S-VLP formulations. Anti-S geometric mean titers (GMT) at day 28 from naïve and treated groups were determined by ELISA. **b\*** Quantification of spike-specific T cells. Isolated splenocytes from vaccinated mice with S-VLP formulations were co-incubated with a spike peptide library. Quantification of IFN-γ+ secreting cells was performed by ELISpot and results were expressed as the mean of IFN-γ +/10$^6$ splenocytes. **c\*\*** Cell-based (hACE2-HEK293T) surrogate neutralization assay. The percentage neutralization means for SARS-CoV-2 was determined from dilutions of 1:25 in unadjuvanted and Adju-phos or 1:75 in AS01b conditions. **a–c** Data are presented as geometric mean ± SEM (10 mice per group). Black dots represent naïve animals, blue for unadjuvanted, red for Adju-phos, and green for AS01b S-VLPs. **d** Body weight change in immunized hamsters infected with SARS-CoV-2 (ancestral strain). The line graph represents daily body weight measurements of naïve and vaccinated animals post-challenge. **e\*\*** Virus quantification by plaque assay. Plaque forming units (PFUs) were quantified in Vero E6 cells infected with hamster lung homogenates. Virus load (log10) is displayed in PFU per gram of lung. **f\*\*** Viral RNA quantification in hamster lungs. RNA samples from naïve and vaccinated animals at five days post-challenge were subjected to viral RNA detection by RT-qPCR. SARS-CoV-2 RNA concentration is displayed by gRNA copies/lung (log10). **d–f** Data are presented as mean ± SEM (6 animals per group). Black dots (gray bars) represent naïve animals, blue is unadjuvanted S-VLPs, orange AS01b and green represents S-VLPs+AS01b. **\***A one-way ANOVA with Tukey's or **\*\***Dunnette's multiple comparison tests was performed to assess significance. $p \leq 0.0001$, $p \leq 0.001$, $p \leq 0.01$ and $p \leq 0.05$ are represented by four, three, two, and one asterisks, respectively, and $p > 0.05$ as not significant (NS).

Differences in viral RNA levels were less pronounced, with a remarkable difference only in the VLPs + AS01b group ($p = 0.015$) (Fig. 4f), likely due to differences in RNA and virus shedding/clearance kinetics as reported in vaccinated and unvaccinated cohorts[49]. Thus, in vivo, models confirmed that S-VLPs+AS01b elicited a strong Th1/Th2 response and fully protected animals against SARS-CoV-2 infection, even at low-nanogram dose levels.

Taken together, these results suggest S-VLPs + AS01b could be more immunogenic than most VLP vaccines currently in preclinical or clinical phases. VBI-2902a used 0.2 and 1 µg doses of adjuvanted S protein from Gag-S-based VLPs to induce neutralizing antibodies and grant protection in murine models, respectively[9]. VLPs generated by the assembly of 4 structural proteins (S-M-E-N) of SARS-CoV-2 need 2.83 µg of adjuvanted antigen to produce neutralizing antibodies (EC50) in mice[4]. ABNCoV2 using split-protein Tag/Catcher technology, where S RBD produced in insect cells is linked to a bacteriophage AP205 capsid-like particle produced in *E. coli*, induces neutralizing activity in preclinical and clinical trials using 6.5 or ≥ 6 µg of adjuvanted antigen, respectively[50,51]. Moreover, effective vaccine dosage in humans of Medicago-Covifenz required a prime dose of 3.75 µg and a similar booster dose of VLP-S[10]. Remarkably, our study indicates that in mouse and hamster models, S-VLPs require as low as 0.06–0.3 µg dose of S protein when adjuvanted with AS01b to induce potent neutralizing activity along with concomitant protection against SARS-CoV-2 challenge.

In summary, in contrast to previously published work using other cell-based expression platforms[4,9,10,27,50–52], we have discovered that expression of SARS-CoV-2 S alone in CHO cells is sufficient to induce assembly and release of abundant, high-density S-coated VLPs. Compared to alternative VLP technologies, S-VLPs not only show similar or better in vivo potency but also permit simplified upstream production and downstream purification processes, which should facilitate low-cost, large-scale manufacturing. We expect that CHO-produced S-VLPs could provide a cost-effective and agile vaccine platform to continue combating the current pandemic including future SARS-CoV-2 variants of concern.

## Data availability

VLP coding sequences of M, E, and S used in this work are available through GenBank (NCBI Reference Sequence: NC_045512.2). Mutations in the S protein sequence for the generation of S-VLPs are fully described in the "Methods" section. Supplementary information contains the following details. Supplementary Fig. 1: Generation of different S-VLP types. Supplementary Fig. 2: Quantification and size determination of S-VLPs by NTA. Supplementary Fig. 3: S protein density estimation on the particle surface and S glycosylation analysis. Supplementary Fig. 4: Cell-based (Vero E6) surrogate neutralization assay for SARS-CoV-2 variants. Supplementary Fig. 5: Digital uncropped images of gels and blots. In addition, the supplementary information file contains additional methods used in the present manuscript. Data generated for glycosylation studies are provided in Supplementary Data File 1. The numerical data underlying Figs. 2b, f, g, 3b, and 4 can be found in Supplementary Data File 2. All other data would be available upon reasonable request to the corresponding author.

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

## Acknowledgements
We thank Dr. Warren Kett at Avitide Inc. for providing the AVIPure-COV2S VLP affinity resin. This is an NRC publication #NRC-HHT_53654A.

## Author contributions
S.P.A.-L., M.S., and Y.D. conceived the study. Cloning, expression, purification, SDS–PAGE, Western blot, and ELISA of VLPs were performed by S.P.A.-L., J.-S.M., M.S., B.C. and J.B. Biophysical characterization of VLPs (NTA, EM, spike modeling on VLP surface, etc.) was performed by S.H., S.P.A.-L., D.Z. J.-Y.C., K.C., and X.W. Spike VLP glycosylation analyses were performed by A.R., A.H., A.S., and J.K. Animal studies were performed and analyzed by B.A., T.M.R., R.D., M.J.M., D.D., and A.T. The manuscript was written by S.P.A.-L., M.S., and Y.D. and reviewed by S.P.A.-L., M.S., Y.D., B.A., and T.M.R.

## Competing interests
S.P.A.-L., M.S., and Y.D. declare filing of a provisional patent application (patent name: ENVELOPED VIRUS LIKE PARTICLES COMPRISING SARS-COV-2 S PROTEIN, Application Number: PCT/IB2023/057279, Status of application: Patent Cooperation Treaty (PCT) request, geographical region: request to PCT for international application). The present manuscript describes the use of S-VLPs as a vaccine candidate against SARS-CoV-2. The remaining authors declare no competing interests.
