## [Peer Review File · Communications Medicine]

Reviewers' comments:

Reviewer #1 (Remarks to the Author):

The authors described a practical platform to produce SARS-CoV-2 Virus like Particles (VLPs) in Chinese Hamster Ovary (CHO) cells. and they showed that their manufactured VLPs had high particle yields, spike density, human-like spike protein glycosylation. Finally, the VLPs can elicit protective humoral and cellular immune responses at low dose levels in vaccinated animal models. Overall, the work was straight-forward and the conclusion was clear. Some concerns were listed below

- 1) In the mice experiments, serum from the VLPs+AS01b group can effectively neutralize the Delta, Beta strains (Fig.4c and Supplementary Fig. 4). Dose the VLPs have neutralization effect against Omicron strain?
- 2) The authors showed VLPs+AS01b can provide protection for immunized hamsters infected with SARS-CoV-2 (ancestral strain) (Fig.4d-f). However, whether VLPs+AS01b can grant similar protection when other SARS-CoV-2 variant infection, particularly against the current pandemic Omicron stain?
- 3) The authors evaluated VLPs could elicit protective humoral and cellular immune responses in rodents (Fig.4). Total anti-spike IgG, IFN- γ + secreting cells and surrogate neutralization assay were conducted in vaccinated mice (Fig.4a-c), while body weight change and virus quantification in immunized hamsters infected with SARS-CoV-2 (Fig.4d-f). Why to change the models when challenge SARS-CoV-2? Whether the VLP had protection in the K18-hACE2 mice, which represents the severe infection of the SARS-CoV-2.
- 4) The TEM quality looks not good, are there some debris or residual iodixanol cushion which presented strong signals in TEM image (Fig.1b and 2e)?

Reviewer #2 (Remarks to the Author):

The study "preclinical evaluation od manufacturable SARS-CoV-2 spike virus like particles produced in CHO cells" of Alpuche-Lazcano et al shows the very interesting development of another promising vaccine candidate against SARS-CoV-2. In contrast to prior studies S protein alone was sufficient to produce VLP in CHO cells. Notably modified CHO cells were used, that did not produce ERVLPs. However, as affinity purification was used later on, I am not convinced that this CHO cell line is required. Yet, as production levels were similar to the standard CHO cell line it also does not negatively influence the production. The formed VLPs were analysed in detail using TEM, Cryo-EM; NTA and Glycan analysis. The authors claim a high titer of their VLPs (Line 305) – but do not put it in context to published production rates in other systems (at least not to yields of S+M+E VLP in eg HEK or insect cells). The authors point out the higher Spike density and thereby high efficiency of there vaccine. In addition, first in vivo data in mice and hamster models were obtained and are quite convincing. The authors demonstrated, that expression of variants (beta,delta,omicron) is also possible and it will be interesting to see in vivo data of the omicron version.

In total it is a convincing study about a new VLP vaccine candidate in CHO cells and of high value for the research community. It is well written and I recommend publishing.

Only a few minor things:

Line 112: I am wondering why you decided for the stabilized version of Spike- could you please elaborate on that? I would assume the wildtype variant to be more realistic and therefore raise more neutralizing antibodies? The Furin site is already cleaved during the production of "real" SARS-

CoV-2 virus (e.g. Hossain et al PMID: 34936124) and virus entry is more efficient when Furin site is not mutated.

Line 136: You write pTT5-Membrane-Envelope protein. Please clarify why M and E protein are encoded on one expression (reference to supplement is missing) plasmid as IRES is probably leading to less production of E protein. Did you test different ratios of the expression vectors?

Line 141: As you are working with the supernatants I won't expect a high DNA contamination, so I am wondering why you are using DNase and if the cell viability was a problem.

Line 486 table 2: 488 is overlaying median size in this pdf version.

Supplementary Figure 2: Quality of a, b and c needs to be improved (black background in the pdf)

Reviewer #3 (Remarks to the Author):

Alpuche-Lazcano et al. reported that the Spike protein of SARS-CoV-2 was expressed using CHO cells (CHO C2) with gag gene disrupted. Their results showed that the CHO cells could generate VLPs derived from S protein. The density of corona-spike on the surface of the s-VLPs is higher than that on the authentic SARS-CoV-2 virus. Interestingly, the authors also demonstrated the S protein is properly glycosylated and could produce significant humoral and cellular immunity. Overall, the experiments were well designed. Important experiments such as antibody neutralisation assay, anti-S determination glycosylation analysis, animal challenge, lung viral RNA detection, and plaque assay were performed. However, there are several concerns that need to be addressed before the paper can be accepted for publication.

Major concerns:

1. Following the purification of the S-VLPs, an antigenicity assay (eg. ELISA with anti-S) should be performed to ensure that the immunodominant region (eg RBD) is properly exposed to the surface of the VLPs.
2. Although the authors determined the amount of the purified S-VLPs, the purity of the VLPs, however, was not reported. It is crucial to ensure the protein is sufficiently pure before it is immunised into animal models.
3. It would be interesting to include experiments such as immunophenotyping and cytokine profile analysis.
4. Please provide an explanation why transgenic mice were not used in the animal challenge study? What are the advantages of using hamster?
5. Figure 1e, authors are required to label micrographs appropriately. The morphology of these virus particles is significantly different particularly the spike conformation and their intensities, as well as their internal density. Authors should provide explanation of these differences or provide better micrographs representing these VLPs.

Minor comments:

1. Figure 1c can be placed in supplementary materials. It is not necessary to be included in the main text.
2. There is a mistake in Table 2 (please check).

June 06th, 2023

Dear Reviewers,

Please find attached our manuscript entitled “Preclinical evaluation of manufacturable SARS-CoV-2 spike virus-like particles produced in CHO cells” by S. Alpuche-Lazcano et al, to be considered for publication in Communications Medicine. In this revised version of our last submitted manuscript, we have addressed your comments by performing additional experiments and providing new data that has been added to our article. Here, in this rebuttal letter, you will find point-by-point answers to your questions and concerns. We hope this new version along with our answers to your comments/concerns can fulfill the publication criteria of communications medicine.

Last but not least, we thank the reviewers for the feedback and invested time that have provided to our manuscript.

Sincerely,

Yves Durocher, PhD.
Principal Research Officer
National Research Council Canada
6100 Royalmount, QC, Canada. H4P 2R2

Referee expertise:

Referee #1: SARS-CoV-2 vaccine production, CHO cells, mouse models, nanoparticle vaccines

Referee #2: Virus-like particles, SARS-CoV-2 vaccine development

Referee #3: Vaccine development, virus-like particles

Reviewers' comments and **authors answers**.

Reviewer #1 (Remarks to the Author):

The authors described a practical platform to produce SARS-CoV-2 Virus like Particles (VLPs) in Chinese Hamster Ovary (CHO) cells. and they showed that their manufactured VLPs had high particle yields, spike density, human-like spike protein glycosylation. Finally, the VLPs can elicit protective humoral and cellular immune responses at low dose levels in vaccinated animal models. Overall, the work was straight-forward and the conclusion was clear. Some concerns were listed below

1) In the mice experiments, serum from the VLPs+AS01b group can effectively neutralize the Delta, Beta strains (Fig.4c and Supplementary Fig. 4). Dose the VLPs have neutralization effect against Omicron strain?

In response to this comment, we have performed a new surrogate neutralization assay to evaluate the response against Omicron BA.4-5 using samples from the previous mouse study. Indeed, there is neutralization activity. The results are now included in the main manuscript (supplementary Fig 4D).

2) The authors showed VLPs+AS01b can provide protection for immunized hamsters infected with SARS-CoV-2 (ancestral strain) (Fig.4d-f). However, whether VLPs+AS01b can grant similar protection when other SARS-CoV-2 variant infection, particularly against the current pandemic Omicron stain?

The surrogate neutralization assay for mice vaccinated with S-VLP formulations showed antibody-neutralizing activity against SARS-CoV-2 (reference strain), especially the S-VLP/AS01b condition, suggesting possible protection in hamsters. We confirmed that S-VLP/AS01b could successfully protect hamsters against the reference strain virus as well. In response to the reviewer's question, we tested pre-challenge serum samples from the hamster study using the same surrogate neutralization assay used for the mouse experiments with reference-strain and BA.4/5 spike proteins. Like for the mice, as shown in Figure 1 (below), serum from reference-strain S-VLP-immunized hamsters (AS01b-adjuvanted) is able to neutralize BA.4/5 spike, although less well than reference-strain spike. Although we have not tested for protection against Omicron virus infection in hamsters, the neutralizing activity of the hamster serum might suggest that S-VLP/AS01b would also be protective against Omicron. Because the mouse and hamster results with BA.4/5 are similar, we propose to include only the mouse results in the manuscript.

Fig 1. Immunization with adjuvanted S-VLPs induces SARS-CoV-2-neutralizing responses against SARS-CoV-2 reference and BA.4-5 strains in Hamster models. Cell-based (hACE2-Vero E6) surrogate neutralization assay. The percentage neutralization mean for SARS-CoV-2 was determined from dilutions of 1:75 in Naïve, S-VLP, AS01b, and S-VLP/AS01b conditions. On the left, plotted data of reference strain. On the right, plotted data of Omicron BA.4-5 strain. Data are presented as mean \pm SEM (6 animals per group). Black dots represent naïve animals, blue is unadjuvanted S-VLPs, orange AS01b and the green represents S-VLPs+AS01b. A one-way ANOVA with Dunnett's multiple comparison tests was performed to assess significance. $p < 0.0001$ is represented by four asterisks whereas $p > 0.05$ as NS.

3) The authors evaluated VLPs could elicit protective humoral and cellular immune responses in rodents (Fig.4). Total anti-spike IgG, IFN- γ + secreting cells and surrogate neutralization assay were conducted in vaccinated mice (Fig.4a-c), while body weight change and virus quantification in immunized hamsters infected with SARS-CoV-2 (Fig.4d-f). Why to change the models when challenge SARS-CoV-2? Whether the VLP had protection in the K18-hACE2 mice, which represents the severe infection of the SARS-CoV-2.

While K18-hACE2 mice are an alternative model for SARS-CoV-2 infection, the disease pathology in this model does not accurately reflect those observed in human patients. The lethality observed in the K18-hACE2 mouse model stems mainly from severe neurological pathology with high viral burden in brain tissues, and not from a respiratory one, something that is rarely seen in severe human cases. The Golden Syrian hamster is currently the most accepted model that better mirrors disease progression and pathology observed in human patients. The main limitation of using the hamster model is the lack of some hamster-specific reagents (antibodies, etc) that narrows its use to vaccine efficacy determination. For this reason, the immunological aspects of the study were carried out in mice.

4) The TEM quality looks not good, are there some debris or residual iodixanol cushion which presented strong signals in TEM image (Fig.1b and 2e)?

Yes, there is normally more debris in VLP samples purified using the iodixanol cushion method (the 15% iodixanol does not separate VLPs from other debris with the same sedimentation coefficient present in CHO supernatants).

Especially for Fig 1b, the production of VLPs was inefficient so the amount of debris relative to VLPs is high.

S-VLP images generated after affinity purification (eg. Fig 2d-e) show much lower amounts of debris which is also confirmed by total protein staining where host cell proteins were drastically reduced compared to iodixanol-sedimented particles (Fig 2c). Our intention with Fig 2e (now Fig 2d zoomed-in image) is to show the reader the abundance of S-VLPs after purification. The low debris content observed in Fig 2d is derived from VLPs (mock purification does not contain debris) but this is unavoidable since this debris are generated during sample preparation for EM staining (please refer to negative-stain transmission electron microscopy (TEM) in the materials and methods section). The TEM quality for Fig 2d is good, in our opinion. It is possible that the image resolution was poor in the PDF version– see enlarged version below.

Affinity purified VLPs (Fig 2d in manuscript)

Reviewer #2 (Remarks to the Author):

The study “preclinical evaluation of manufacturable SARS-CoV-2 spike virus like particles produced in CHO cells” of Alpuche-Lazcano et al shows the very interesting development of another promising vaccine candidate against SARS-CoV-2. In contrast to prior studies S protein alone was sufficient to produce VLP in CHO cells. Notably modified CHO cells were used, that

did not produce ERVLPs. However, as affinity purification was used later on, I am not convinced that this CHO cell line is required. Yet, as production levels were similar to the standard CHO cell line it also does not negatively influence the production.

We are indeed able to produce S-VLPs in unmodified CHO cells (Fig 1 from manuscript), but we can demonstrate that engineering cells to prevent contamination with ERVLPs is important for the purity of the final product, even if affinity purification is used.

As shown below (Fig 2), we have tried affinity purification for S-VLPs from non-ERVLP-KO CHO cells and we still observe the presence of ERVLP proteins in the final product by western blotting (compared to CHO-C2 KO cells, where the protein is not detectable).

Fig 2. Representative immunoblot of purified S-VLPs from CHO-55E1 and CHO-C2 cell supernatants. The top panel of the gel was stained with anti-HA (Sigma #6908). The bottom fragment was stained anti-ERVLP-Gag p30.

The formed VLPs were analysed in detail using TEM, Cryo-EM; NTA and Glycan analysis. The authors claim a high titer of their VLPs (Line 305) but do not put it in context to published production rates in other systems (at least not to yields of S+M+E VLP in eg HEK or insect cells).

We have changed the section heading for **“SARS-CoV-2 S expression in CHO cells is sufficient to generate and release VLPs in cell supernatants”** because the previous version suggested that S-VLPs were “abundant” without clearly comparing this platform to other methods. This change would allow readers to discover in the next section **“Purification of SARS-CoV-2 S-VLPs produced by ERVLP-KO CHO cells”** which includes information about S-VLP production and purification yields compared to other platforms including VLPs generated in insect cells. Notably, we would have liked to discuss further VLP technologies in clinical trials like Covifenz

(Medicago) or the S+M+E+N VLP vaccine candidate produced in HEK293 cells by Middle East Technical University (Turkey). However, data about S protein quantification or VLP number in bulk production or purified material is not provided. Information about other SARS-CoV-2 vaccines in preclinical trials such as ExcepGen, S, M, N and E with and without Myxoma virus from Arizona University, IrsiCaixa AIDS Research/IRTA-CReSA/Barcelona Supercomputing Centre/Grifols with their HIV-VLP-S approach, is not publicly available and we cannot provide such comparison.

The authors point out the higher Spike density and thereby high efficiency of their vaccine. In addition, first in vivo data in mice and hamster models were obtained and are quite convincing. The authors demonstrated, that expression of variants (beta,delta,omicron) is also possible and it will be interesting to see in vivo data of the omicron version.

We are currently working on providing new evidence with novel omicron subvariants, but as time goes on, SARS-CoV-2 evolves rapidly (now the frequency of XBB.1.5 and other related Omicron variant sub lineages are increasing in many countries) which causes a cat-and-mouse effect. We are working on novel approaches that would ultimately combine different strains that might protect for a larger array of current and future variants.

To demonstrate our current VLPs can trigger neutralizing activity against omicron subvariants, we have performed new surrogate neutralization assays to evaluate the response against Omicron BA.4-5. These results can be consulted in the main manuscript and are displayed in supplementary Fig 4D.

In addition, Fig 1 in this document demonstrates Omicron BA.4-5 neutralizing activity in hamsters.

In total it is a convincing study about a new VLP vaccine candidate in CHO cells and of high value for the research community. It is well written and I recommend publishing.

Only a few minor things:

Line 112: I am wondering why you decided for the stabilized version of Spike- could you please elaborate on that? I would assume the wildtype variant to be more realistic and therefore raise more neutralizing antibodies? The Furin site is already cleaved during the production of "real" SARS-CoV-2 virus (e.g. Hossain et al PMID: 34936124) and virus entry is more efficient when Furin site is not mutated.

We did try producing VLPs using spike constructs with and without the prefusion-stabilizing and furin-site mutations. VLPs were produced efficiently without the stabilizing mutations but with the furin site intact, VLP formation was greatly reduced. We are not certain why mutating the furin site is required for efficient VLP formation, but to ensure that the full spike ectodomain, including the S1 subunit, is displayed on the VLPs, we opted to keep the furin site mutated for all the productions described in the manuscript. Likewise, although we don't have evidence that spikes displayed on our VLPs could adopt a post-fusion conformation without the stabilizing prolines present, we opted to include these to have more confidence that the desired prefusion conformation would be dominant.

Line 136: You write pTT5-Membrane-Envelope protein. Please clarify why M and E protein are encoded on one expression (reference to supplement is missing) plasmid as IRES is probably leading to less production of E protein. Did you test different ratios of the expression vectors?

The design of this construct was based on a 2008 paper (Siu et al., J Virology, doi:10.1128/JVI.01052-08). These authors concluded that using a bicistronic construct with E downstream of an IRES (and therefore expressed less than M) led to most efficient SARS-CoV-1 VLP production. We did not test other vector designs (or M:E ratios) in our study.

Line 141: As you are working with the supernatants, I won't expect a high DNA contamination, so I am wondering why you are using Denarase and if the cell viability was a problem.

Our method for VLP production is based on transient transfection. Benzonase (Denarase) endonuclease is commonly used to reduce the amount of residual plasmid and cellular genomic DNA and RNA for viral vector purification, most often from supernatants of transiently transfected HEK293 cells (Sastry, L et al., 2004; Nasimuzzaman, Md et al., 2016). We are using a different cell type (CHO) for VLP production, but we retained this step based on our standard viral vector protocol. Cell viability is good on the day of harvest (>95 %), but we never tried omitting the Denarase treatment.

Line 486 table 2: 488 is overlaying median size in this pdf version.

Removed

Supplementary Figure 2: Quality of a,b and c needs to be improved (black background in the pdf)

We have added these figures in BMP File (.bmp) and now our supplementary figures in PDF format look fine.

Unfortunately, we cannot modify the background of NTA plots because the NanoSight NTA 3.2 Analytical Software (Malvern Panalytical) display FTLA and scatterplots with black background without any option to modify them.

Reviewer #3 (Remarks to the Author):

Alpuche-Lazcano et al. reported that the Spike protein of SARS-CoV-2 was expressed using CHO cells (CHO C2) with gag gene disrupted. Their results showed that the CHO cells could generate VLPs derived from S protein. The density of corona-spike on the surface of the s-VLPs is higher than that on the authentic SARS-CoV-2 virus. Interestingly, the authors also demonstrated the S protein is properly glycosylated and could produce significant humoral and cellular immunity. Overall, the experiments were well designed. Important experiments such as antibody neutralisation assay, anti-S determination glycosylation analysis, animal challenge, lung viral RNA detection, and plaque assay were performed. However, there are several concerns that need to be addressed before the paper can be accepted for publication.

Major concerns:

1. Following the purification of the S-VLPs, an antigenicity assay (eg. ELISA with anti-S) should be performed to ensure that the immunodominant region (eg RBD) is properly exposed to the surface of the VLPs.

To answer the reviewer's concern, we have developed an ELISA assay with and without an hACE2 coating; please refer to supplementary methods. The binding of S-VLPs to this receptor followed by the S detection with polyclonal anti-S demonstrates proper RBD exposure at the VLP surface. ELISA results are addressed in the main manuscript and displayed in Fig 2G.

2. Although the authors determined the amount of the purified S-VLPs, the purity of the VLPs, however, was not reported. It is crucial to ensure the protein is sufficiently pure before it is immunised into animal models.

We are not sure that we understand this particular comment/concern. We have demonstrated the purity of VLPs by multiple methods including analysis of total protein by SDS-PAGE (Fig 2C) compared to the reference material (S STD) Smt1 (SMT1-1: SARS-CoV-2 Spike Glycoprotein Reference Material). Our plasmid for VLPs shares the same ectodomain sequence as the Smt1 and we estimated that spike represents >80% of total VLP protein content. Additional determination of host cell proteins (HCPs) by ELISA in purified VLP samples (table 1) showed the purity of VLPs containing <10 % of HCPs. We have also demonstrated, by Cryo-EM (Fig 3a) the resemblance of the spike on VLPs to the ones determined by previous publications. Also, although not quantitative, TEM imaging demonstrated that few non-spike-VLP particles are present in the affinity-purified material.

3. It would be interesting to include experiments such as immunophenotyping and cytokine profile analysis.

We agree with the reviewer that immunophenotyping and cytokine profile would be interesting to explore with the S-VLPs vaccine. In our experience, these signals usually depend mostly on the adjuvant in use and not necessarily the antigen, which is the main focus of this manuscript. We plan to pursue more immunological data related to cytokine profile, immunophenotyping and a more characterized Th response in upcoming work.

4. Please provide an explanation why transgenic mice were not used in the animal challenge study? What are the advantages of using hamster?

See question 3 from Reviewer 1.

5. Figure 1e, authors are required to label micrographs appropriately. The morphology of these virus particles is significantly different particularly the spike conformation and their intensities, as well as their internal density. Authors should provide explanation of these differences or provide better micrographs representing these VLPs.

New micrographs from recent VLP productions have been added to the main manuscript. An enhanced resolution of these micrographs might be sufficient to observe similarities between sedimented S-VLPs of reference and variants (Fig 1c and Fig 1e).

Notably, we have performed TEM several times with iodixanol-purified VLPs, but the image quality is consistently poorer than with affinity-purified samples. VLPs sedimented on an iodixanol cushion by ultracentrifugation contain other particles with a similar sedimentation coefficient. Transmission electron microscopy (TEM) in these samples is quite challenging and also difficult to provide high-resolution images in contrast to purified S-VLPs (Fig 2e, TEM, high resolution Hitachi 7700). After centrifugation, sedimented VLPs are more pleomorphic, not as rounded as those purified by chromatography and sometimes, we can find VLP aggregates that look like one large fused VLP. Therefore, differences between VLPs even from the same sample can be visible in iodixanol-sedimented samples.

Minor comments:

1. Figure 1c can be placed in supplementary materials. It is not necessary to be included in the main text.

We think Fig 1c is important for comparison of sedimented S-VLPs (reference strain) and those produced by M/E/S co-expression (Fig 1b).

2. There is a mistake in Table 2 (please check).

Line 486 in table 2 was removed

REVIEWERS' COMMENTS:

Reviewer #1 (Remarks to the Author):

I have no questions.

Reviewer #2 (Remarks to the Author):

The authors answered all of my questions to my satisfaction. I recommend publishing.

Reviewer #3 (Remarks to the Author):

The authors have addressed all the comments. I have no further comments or suggestions for the manuscript. Therefore, I recommend the manuscript to be accepted for publication.